

# Developmental exposure to the DE-71 mixture of polybrominated diphenyl ether (PBDE) flame retardants induce a complex pattern of endocrine disrupting effects in rats

Louise Ramhøj, Karen Mandrup, Ulla Hass, Terje Svingen and Marta Axelstad

Division of Diet, Disease Prevention and Toxicology, National Food Institute, Technical University of Denmark, Kgs. Lyngby, Denmark

## ABSTRACT

Polybrominated diphenyl ethers (PBDEs) are legacy compounds with continued widespread human exposure. Despite this, developmental toxicity studies of DE-71, a mixture of PBDEs, are scarce and its potential for endocrine disrupting effects in vivo is not well covered. To address this knowledge gap, we carried out a developmental exposure study with DE-71. Pregnant Wistar rat dams were exposed to 0, 40 or 60 mg/kg bodyweight/day from gestation day 7 to postnatal day 16, and both sexes were examined. Developmental exposure affected a range of reproductive toxicity endpoints. Effects were seen for both male and female anogenital distances (AGD), with exposed offspring of either sex displaying around 10% shorter AGD compared to controls. Both absolute and relative prostate weights were markedly reduced in exposed male offspring, with about 40% relative to controls. DE-71 reduced mammary gland outgrowth, especially in male offspring. These developmental *in vivo* effects suggest a complex effect pattern involving anti-androgenic, anti-estrogenic and maybe estrogenic mechanisms depending on tissues and developmental stages. Irrespective of the specific underlying mechanisms, these in vivo results corroborate that DE-71 causes endocrine disrupting effects and raises concern for the effects of PBDE-exposure on human reproductive health, including any potential long-term consequences of disrupted mammary gland development.

## INTRODUCTION

Normal development through gestation and early postnatal life is essential for lifelong health and disruption to developmental processes can result in adverse effects and increased susceptibility to disease. Since the environment where development occurs plays a key role in ensuring proper development, environmental stressors are also major contributors to the development of diseases. This includes endocrine disrupting chemicals, which can disturb molecular and biological pathways that, through their effects on the endocrine

Corresponding author
Marta Axelstad, maap@food.dtu.dk

systems, can change developmental processes and cause adverse health effects (*Gore et al., 2015*).

Polybrominated diphenyl ethers (PBDEs) are legacy compounds that were widely used as flame retardants in various consumer and industrial products. They have been shown to cause numerous adverse health effects in humans and animals and are now restricted under the Stockholm Convention on Persistent Organic Pollutants (*Stockholm Convention, 2019*). Unfortunately, humans are still exposed to PBDEs, as they are persistent in indoor and outdoor environments and bioaccumulate.

Among the many human health effects for which PBDEs have been implicated, thyroid hormone system disruption, neurotoxicity and reproductive disorders are prevalent (*Boas, Feldt-Rasmussen & Main, 2012*; *Attina et al., 2016*; *Trasande et al., 2016*). Still, their specific endocrine mechanisms of action are not well understood and a characterization of the adverse reproductive effects in perinatally exposed rat offspring remain lacking. Even though PBDEs are largely banned, such information is still valuable as human exposure will continue and since it will help us better understand the current disease burden of these legacy chemical substances as well as of endocrine disruptors in general. Perhaps equally important is its applicability towards improvement of current safety assessment regimens, including considerations for mixture effects.

Also, experimental studies have shown various potential endocrine effects of DE-71 and its metabolites. Amongst the effects potentially relevant for reproductive health are weak anti-estrogenic and estrogenic effects *in vitro* and *in vivo* (*Mercado-Feliciano & Bigsby, 2008a*; *Mercado-Feliciano & Bigsby, 2008b*) as well as clear anti-androgenic properties *in vitro* (*Stoker et al., 2005*) and in exposed male rats (*Stoker et al., 2004*; *Stoker et al., 2005*), while the potential effects of DE-71 in a developmental toxicity scenario are understudied. A recent meta-analysis on PBDEs suggest broad adverse reproductive effects in postnatal male rats (*Zhang et al., 2020*). Based on this, we surmised that the DE-71 mixture could reduce androgen signaling during fetal life and induce effects resembling the testicular dysgenesis syndrome (*Skakkebaek et al., 2016*).

To test our hypothesis, and hence contribute new knowledge to the *in vivo* endocrine modes of action of PBDEs, we conducted a developmental rat toxicity study to determine the effects of early life exposure to DE-71 on endocrine-sensitive endpoints that have not previously been investigated or only investigated at lower doses with smaller statistical power. These endpoints were anogenital distance (AGD) (*Schwartz et al., 2019*), nipple retention (NR) (*Schwartz et al., 2021*), and a few selected reproductive postnatal and adult organ weights. We also included analyses of mammary gland development in both male and female offspring by whole-mount assessments. Mammary glands are rarely investigated in reproductive toxicity studies despite their sensitivity to endocrine disruption and importance for reproductive health (*Fenton, Reed & Newbold, 2012*; *Gouesse et al., 2019*). Our study corroborates the endocrine activity of DE-71 *in vivo* and shows how multiple active endocrine mechanisms can result in a complex *in vivo* effect pattern.

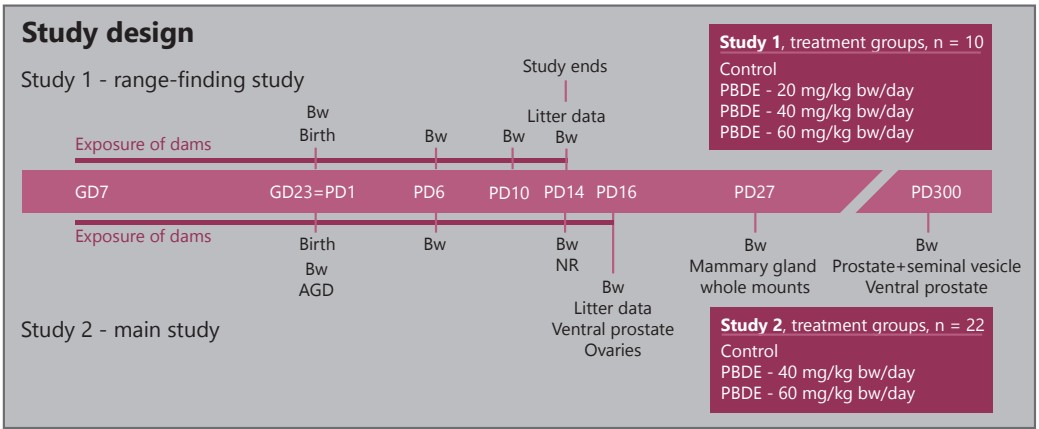

**Figure 1** **Study design for developmental toxicity studies with a technical mixture of brominated flame retardants, DE-71.** Study 1 ($n = 10$ dams) was a range-finding study to choose doses for the larger Study 2 ($n = 22$ dams) that investigated endocrine sensitive endpoints in the perinatally exposed rat offspring. AGD, anogenital distance; Bw, body weight; GD, gestation day; NR, nipple retention; PBDE, polybrominated diphenyl ethers (DE-71); PD, postnatal day.

## MATERIALS AND METHODS

### Chemicals

The test compound was a commercial mixture of PBDEs, DE-71 (pentabrominated diphenyl ethers (pentaBDE) from the Great Lakes Chemical Corporation of West Lafayette, IN, CAS No. 32534-81-9, lot 7550OK20A), a kind gift from Dr. Kevin Crofton at the U.S. Environmental Protection Agency. The manufacturer reports DE-71 to contain 50–62% pentaBDE, 24–38% TetraBDE, 4–12% hexaBDE, and 0–1% triBDE (amounting to a possible total of 154 congeners) (*ENVIRON International Corporation, 2003*). Corn oil (Sigma-Aldrich) was used as control compound and vehicle for all treatments.

### Animals and treatment

We conducted two rat toxicity studies (Fig. 1). Study 1 (range-finding) aimed at determining the highest doses of DE-71 that could be given to pregnant and lactating rat dams without causing overt systemic toxicity in either dams or offspring. Study 2 (main study) aimed at assessing a series of endocrine sensitive endpoints to clarify the full potential endocrine effect pattern of DE-71 in a developmental toxicity study. We also assessed effects on the thyroid hormone system and on neurodevelopment (L. Ramhøj & M. Axelstad, in preparation). In both studies, time-mated nulliparous young adult Wistar rats (HanTac: WH, SPF, Taconic Europe, Ejby, Denmark) were received on gestation day (GD) 3 of pregnancy (day of plug-detection designated GD1), randomly distributed for pairwise housing and on GD4 pseudo-randomly divided into groups with similar weight distribution.

The expected day of delivery, GD23, was designated PD1 irrespective of the actual day of delivery.

In Study 1, dams were divided into four groups ($n = 10$ per group) and exposed to vehicle control or 20, 40 or 60 mg/kg body weight (bw)/day DE-71. Study 2 was performed

in two balanced blocks with 3 groups ($n = 22$ per group, in order to have sufficient sample size for mammary gland examinations) exposed to vehicle, 40 or 60 mg/kg bw/day DE-71. Dosing of the dams was performed daily by gavage in the morning from GD7 to GD22, and again from postnatal day (PD)2 –PD14 in Study 1 (study termination) and PD2-16 in Study 2. Dosing was performed with a metal gastric tube with either vehicle (corn oil) or experimental solutions at a constant volume of 2 ml/kg/day, with the individual doses based on the body weight of the animal prior to dosing.

Dams were housed pairwise until GD17 and individually thereafter. We used semitransparent plastic cages ($15 \times 27 \times 43$ cm) with aspen wood hides and with aspen bedding (Tapvei, Gentofte, Denmark) placed on racks (balanced for treatment group) in an animal room with controlled environmental conditions: reversed light/dark cycles of 12 h (light from 9 pm-9 am, light intensity 500 lux), temperature $22 \pm 1$ °C, humidity $55 \pm 5\%$, and ventilation changing air 10 times per hour. All animals were fed ad libitum on a standard diet, Altromin 1314 (soy- and alfalfa-free; Altromin GMbH, Lage, Germany), and acidified tap water was provided ad libitum in PSU bottles (84-ACBT0702SU Techniplast).

All endpoint examination procedures were carried out with treatment groups represented in a random order (*i.e.,* not examining all controls first, then treated animals). Examinations and outcome measures of live animals were conducted blinded to exposure group while data analysis was conducted unblinded.

Animals were observed twice daily for health and signs of overt toxicity (such as hunched posture and raised fur). Criteria followed for euthanasia of animals: if it was considered irresponsible or unethical to let the animal live or if they were in a severe condition (expected to die within 24hrs). Accordingly, some litters and pups in Study 1 (see Table 1) were euthanized, as the dams did not take care of them and they were without milk in their stomachs. This phenomenon only took place in Study 1 and was unrelated to exposure, it also includes some dams that cannibalized their pups. In Study 2, one dam was euthanized on GD23 as she appeared unwell and incapable of giving birth (necropsy did not reveal any explanations for the dystocia).

Animal experiments were carried out at the DTU National Food Institute (Mørkhøj, Denmark) facilities. Ethical approval was obtained from the Danish Animal Experiments Inspectorate, with authorization number 2012-15-2934-00089 C4. The experiments were overseen by the National Food Institute's in-house Animal Welfare Committee for animal care and use. All methods in the study were performed in accordance with relevant guidelines and regulations.

## Birth and postnatal development

On the morning after overnight birth, dam and pup body weights were registered, and the pups were sexed and checked for macroscopic anomalies. Body weights of offspring were recorded on PD6, PD10 and PD14 in Study 1 and on PD6, PD14 and PD27 in Study 2.

Endocrine sensitive endpoints were investigated in Study 2. Anogenital distance (AGD), the distance between the anus and the genital papilla was measured in all live offspring on PD1, using a stereomicroscope with a micrometer eyepiece. The AGD index (AGDi) was calculated by dividing the AGD with the cube root of the body weight. On PD14

**Table 1  Pregnancy and litter data: Study 1.**

| Dams and litters | Control | 20 mg/kg DE-71 | 40 mg/kg DE-71 | 60 mg/kg DE-71 |
|---|---|---|---|---|
| No. of dams (viable litters) | $n = 10$ (8) | $n = 10$ (7) | $n = 10$ (8) | $n = 10$ (8) |
| **Dam body weight gain** | | | | |
| Dam bw-gain, GD7-GD21 (g) | 79.6 ± 13.0 | 87.1 ± 8.4 | 82.2 ± 14.6 | 90 ± 17.2 |
| Dam bw-gain, GD7-PD1 (g) | 16.5 ± 8.8 | 8.9 ± 20.0 | 2.1 ± 10.9 | 4.5 ± 9.2 |
| Dam bw-gain, PD1-14 (g) | 15.6 ± 13.4 | 16.6 ± 14.3 | 26.1 ± 10.2 | 28.4 ± 10.8 |
| **Litters** | | | | |
| Gestation length | 22.9 ± 0.6 | 23.0 ± 0.0 | 23.0 ± 0.0 | 23.0 ± 0.0 |
| Postimplantation loss (prenatal mortality) (%) | 16.0 ± 12.6 | 6.0 ± 5.0 | 4.7 ± 5.5 | 4.9 ± 6.4 |
| Perinatal loss[a] (pre- and postnatal mortality) (%) | 32.3 ± 37.7 | 21.0 ± 32.0 | 6.6 ± 6.5 | 27.6 ± 39.1 |
| Litter size (no.) | 9.6 ± 4.3 | 11.3 ± 2.5 | 11.8 ± 3.4 | 11.8 ± 3.4 |
| Postnatal death[a] (%) | 21.0 ± 41.8 | 15.1 ± 34.7 | 2.0 ± 3.7 | 23.5 ± 13.0 |
| Sex ratio, males/females (%) | 58 /42 ± 17.9 | 56 / 44 ± 11.7 | 46 / 54 ± 12.9 | 52 / 48 ± 9.2 |
| **Offspring** | | | | |
| Male birth weight (g) | 6.2 ± 0.7 | 6.1 ± 0.9 | 6.2 ± 0.9 | 5.8 ± 0.7 |
| Female birth weight (g) | 5.9 ± 0.6 | 5.9 ± 0.6 | 5.9 ± 0.7 | 5.5 ± 0.7 |
| Mean bw-gain PD1-6 (g) | 7.4 ± 1.2 | 6.9 ± 1.7 | 6.0 ± 1.3 | 5.9 ± 1.5 |
| Mean bw-gain PD6-14 (g) | 23.1 ± 2.0 | 24.2 ± 1.0 | 21.7 ± 1.3 | 22.1 ± 1.8 |

**Notes.**

bw, body weight; GD, Gestation day; PD, postnatal day.

[a]Unrelated to exposure there were dams cannibalizing or not taking care of their pups in this study.

Data represented as Mean ± SD.

all offspring had their areolas/nipples counted. Nipple retention (NR) of male pups was defined as the number of areolas/nipples (a dark focal area with or without a nipple bud) visible where nipples are usually located (along the milk lines) in female pups (*Schwartz et al., 2021*). Both AGD and NR were assessed by the same technician, blinded to exposure group.

On PD27, one male pup from each litter was weaned and housed pairwise with another male from the same group (when no other males were available, housing was with a littermate).

## Necropsy

Before necropsy was performed, all animals were weighed, anesthetized with $CO_2/O_2$ and killed by decapitation. Selected reproductive organs were examined in offspring killed on PD16, PD27 and PD300 in Study 2. On PD16 one male and one female from each litter were killed and ovaries and ventral prostate were excised and weighed. Mammary glands were collected as whole-mounts from one male and one female on PD27. On PD300, one male offspring from each litter was terminated and prostate together with seminal vesicle was excised and weighed. The ventral prostate was subsequently dissected and weighed alone.

All dams not giving birth (*i.e.,* with resorbed embryos or never pregnant) were humanely killed on PD3, those with litters were humanely killed, along with remaining pups, on PD14

in Study 1 and on PD27 in Study 2. Implantation scars in uteri were counted to determine pregnancy rates and resorptions.

## Whole-mounts

Mammary glands were excised and spread onto a glass slide, covered with parafilm and placed under pressure for 2 h. The whole-mounts were subsequently fixed in 10% neutral buffered formalin, stained with alum carmine, dehydrated in alcohol and cleared with xylene. The slides were scanned in a flatbed scanner at 4800 dpi and images of the 4th gland evaluated as described in *Mandrup et al., (2015)* for outer area (defined as the smallest polygon enclosing the gland), transverse growth (as defined by *Mandrup et al. (2012)*), longitudinal growth, distance to lymph node, distance to 5th gland ($n = 13$–17, except distance to the 5th gland where $n = 11$, 4 and 10 in control, PBDE-40 and PBDE-60, respectively) and number of terminal end buds (TEB) (TEB was defined as tear-drop shaped buds with a diameter $>100$ µm in zone C, as defined by (*Russo & Russo, 1996*) ($n = 17$, 11 and 14 in control, PBDE-40 and PBDE-60, respectively). Distance to lymph node was measured and scored (scores 1–3, with a score of 1 representing mammary tissue not reaching the lymph node, score 2 representing mammary tissue reaching the lymph node and score 3 was given to mammary glands where the tissue reaches beyond the lymph node). All data were assessed for males and females separately and blinded to exposure. In addition, for area measurements, longitudinal and transverse growth data from both sexes were pooled for analysis with increased statistical power. Images were analyzed with Image Pro Plus 7.0 (Media Cybernetics, Bethesda, MD, USA) and calibration performed for each picture before measurements were made.

## Statistical analysis

The alpha level for statistical significance was always set to 5% and all samples were included in the analysis. Data with normal distribution and homogeneity of variance were analyzed for treatment-related effect differences relative to the control by analysis of variance (ANOVA) followed by Dunnet's correction for multiple comparisons. Data were transformed if these conditions were not met and data not fulfilling the criteria were analyzed using a non-parametric statistical test (Kruskal–Wallis with Dunn's multiple comparison test). When relevant, body weight was included as a covariate in the analysis (ANCOVA), *e.g.*, for terminal organ weights and whole-mount measurements. Litter effects were accounted for by only analyzing one pup per litter, using litter means (offspring growth) or by including the litter as a random effect variable (AGD, NR and pooled whole mounts). Distance to lymph node was evaluated as the distance and as scores. The scores were analyzed using Kruskal–Wallis and using a two-sided Fishers exact test to compare the number of animals with glands reaching past the lymph node with the number not reaching past. The number of nipples/areolas (NR) was assumed to follow a binomial-distribution with a response range between 0 and 12 (12 assumed to reflect the biologically possible maximal number of nipples in rats). Litter effects on NR and over-dispersion in the data were accounted for by using Generalized Estimating Equations (GEE) as reported in (*Christiansen et al., 2012*).

SAS Enterprise guide 4.3 (2010) (SAS Institute Inc, Cary, NC, USA) and GraphPad Prism 5 (Graphpad Software, San Diego, CA; USA) was used for statistical analysis.

## RESULTS

### Effects on pregnancy, postnatal growth and general toxicity

We administered DE-71 at doses of 20, 40 and 60 mg/kg to the pregnant and lactating dams from GD7 to PD14/16. Pregnancy and litter data from Study 1 and Study 2 are listed in Tables 1 and 2. In Study 1, there were signs of decreased maternal weight gain at 40 and 60 mg/kg. Pups from the 60 mg/kg bw/day exposure group weighed 8% less than control pups at birth and gained 20% less during the first postnatal week. However, none of these findings were statistically significant. Thus, 60 mg/kg was chosen as the highest dose in Study 2 as it did not induce excessive systemic toxicity in dams and pups.

In Study 2, dam weight gain during pregnancy was significantly decreased at 60 mg/kg (Table 2). The high dose pups had decreased body weight on PD6 (Table 2) and body weight remained non-significant reduced throughout the postnatal period (Table 2) with non-significant reductions also on PD27 ($\sim$7% reduction). Gestation lengths, litter size and pup mortality was not affected by exposure.

### Effects on markers of early-life endocrine disruption

A shorter anogenital distance (AGD) and retained nipples in male offspring are considered sensitive markers of disrupted androgen action during development (*Schwartz et al., 2019*). DE-71 exposure induced a significantly shorter AGD and smaller AGDindex (AGDi, AGD adjusted for pup body size) at both 40 and 60 mg/kg. However, the dose–response curve was 'flat' in that reductions in AGD/AGDi were comparable between the two dose groups (10/11% reduced in the low dose group and 8/6% in the high dose group). The effect on AGD was also comparable between both sexes (Fig. 2A and Table 2). Nipple retention was increased in males from the high dose group ($p = 0.0249$). Notably, the mean number of 0.25 nipple in these high dose males is considered very small (Fig. 2B) and falls well within the range of our historic control data (*Schwartz et al., 2021*).

### Early- and late-life effects on reproductive tissue weights

Weights of selected reproductive organs were assessed both early in life and in adulthood. DE-71 at a dose of 60 mg/kg reduced absolute and relative male PD16 prostate weights with 37% and 42%, respectively (Fig. 3 and Table 3). In female offspring, ovary weights were assessed on PD16 and showed no significant differences between controls and exposed animals.

On PD300, changes to prostate weights in exposed offspring were no longer apparent, nor were effects on seminal vesicle or ventral prostate weights (Table 3).

### Effects on mammary gland development

We assessed mammary gland development by whole-mounts in both male and female offspring at PD27 and found that mammary gland outgrowth was stunted by perinatal exposure to DE-71. In male offspring, the transverse growth and mammary gland areas

**Table 2   Pregnancy and litter data: Study 2.**

| Dams and litters | Control | 40 mg/kg DE-71 | 60 mg/kg DE-71 |
|---|---|---|---|
| No. of dams (viable litters) | $n = 22$ (21) | $n = 22$ (20) | $n = 22$ (20) |
| *Dam body weight gain* | | | |
| Dam, GD7-GD21 (g) | 84.5 ± 13.6 | 83.8 ± 14.2 | 78.6 ± 16.6 |
| Dam, GD7-PD1 (g) | 13.0 ± 11.1 | 11.8 ± 12.5 | **2.8 ± 11.0**[*] |
| Dam, PD1-14 (g) | 42.4 ± 13.1 | 37.0 ± 11.4 | 39.9 ± 17.8 |
| Dam, PD14-27 (g) | −20.9 ± 11.6 | −21.5 ± 9.6 | **−12.0 ± 10.9**[*] |
| *Litters* | | | |
| Gestation length | 23.0 ± 0.3 | 23.0 ± 0.2 | 23.0 ± 0.1 |
| Postimplantation loss (prenatal mortality) (%) | 6.6 ± 8.6 | 6.3 ± 7.9 | 11.7 ± 23.9 |
| Perinatal loss[a] (pre- and postnatal mortality) (%) | 10.3 ± 8.3 | 7.1 ± 8.0 | 21.2 ± 31.9 |
| Litter size (no.) | 10.9 ± 3.0 | 10.7 ± 3.1 | 10.8 ± 3.6 |
| Postnatal death (%) | 3.8 ± 6.1 | 0.8 ± 2.5 | 9.5 ± 26.0 |
| Sex ratio, males/females (%) | 45/55 ± 14 | 46/54 ± 16 | 46/54 ± 14 |
| *Offspring* | | | |
| Male birth weight (g) | 6.2 ± 0.4 | 6.3 ± 0.7 | 5.9 ± 0.6 |
| Female birth weight (g) | 5.9 ± 0.5 | 6.0 ± 0.6 | 5.8 ± 0.7 |
| Male AGDi | 2.13 ± 0.06 | **1.90 ± 0.09**[***] | **1.99 ± 0.08**[***] |
| Female AGDi | 1.15 ± 0.04 | **1.02 ± 0.03**[***] | **1.09 ± 0.05**[***] |
| Male bw PD6 (g) | 12.5 ± 1.8 | 12.3 ± 2.7 | **11.1 ± 2.1**[**] |
| Female bw PD6 (g) | 12.2 ± 1.9 | 12.0 ± 2.3 | **11.1 ± 2.3**[**] |
| Male bw PD14 (g) | 27.9 ± 4.1 | 28.1 ± 5.9 | 25.6 ± 4.4 |
| Female bw PD14 (g) | 27.3 ± 4.4 | 27.5 ± 5.5 | 25.8 ± 5.2 |
| Male bw PD27 (g) | 65.3 ± 1.7 | 65.1 ± 1.9 | 61.2 ± 1.6 |
| Female bw PD27 (g) | 67.2 ± 1.7 | 67.5 ± 2.5 | 62.0 ± 1.8 |

**Notes.**

[*]$p < 0.05$.
[**]$p < 0.01$.
[***]$p < 0.001$.

bw, body weight; GD, Gestation day; PD, postnatal day.

Data shown as Mean ± SD.

were statistically significantly reduced in the highest exposure group (Fig. 4). Females appeared to display a similar trend, albeit not statistically significant. When analyzing pooled data from both sexes (litter effects were accounted for by including the litter as a random nested factor in the statistical analysis) there was an effect on area in both dose groups and on the transverse growth in the high dose group (Fig. 4). The observed effects are likely not due to reduced body weight of the offspring in the high dose as the effects were observed both on absolute numbers and when accounting for the body weight of the animals in the statistical analysis.

## DISCUSSION

While production and use of DE-71 is no longer permitted, the constituents of the DE-71 mixture still account for a significant proportion of the brominated flame retardants found

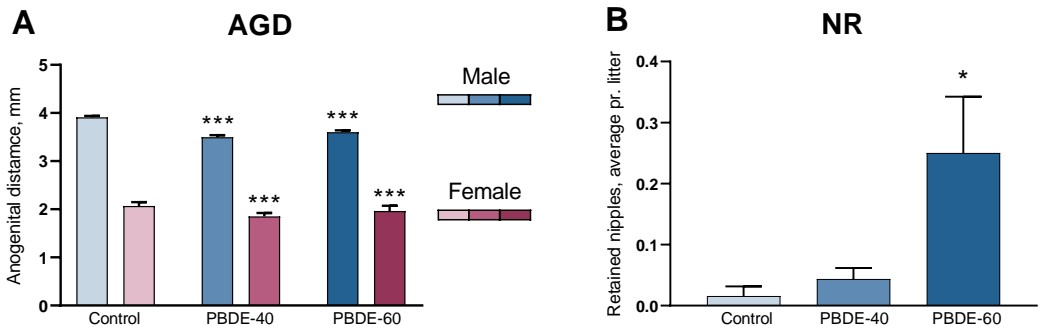

**Figure 2** **Anogenital distance (AGD) and nipple retention (NR) in rat offspring after perinatal exposure to brominated flame retardants (DE-71).** (A) Reduced AGD in PD1 male and female offspring. (B) NR in male PD14 offspring. $n = 19$–21 litters. Statistical analysis performed on all pups from each litter with adjustment for litter effects. Litter means+SEM, *$p < 0.05$ and ***$p < 0.001$ compared to control.

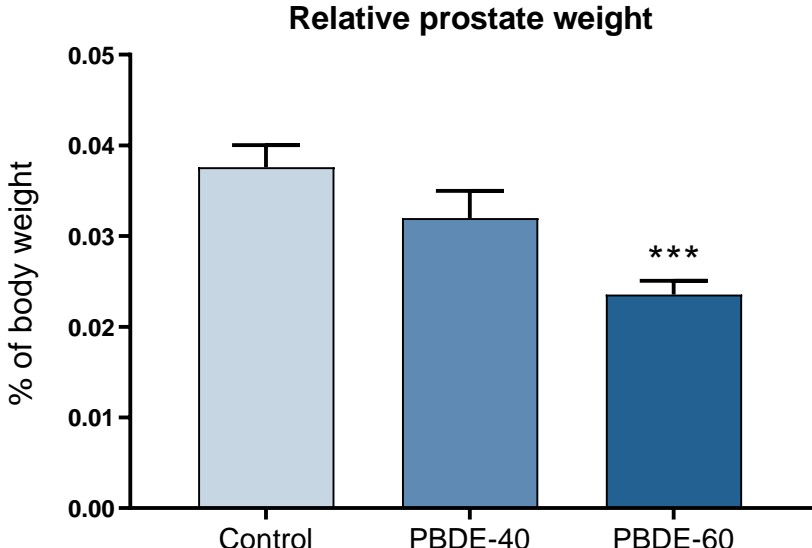

**Figure 3** **Relative ventral prostate weights in PD16 male offspring after perinatal exposure to brominated flame retardants (DE-71).** Mean+SEM, $n = 16$–19. ***$p < 0.001$.

in house dust (*Bramwell et al., 2016*). Humans, including babies and toddlers, continue to be exposed to PBDEs (*Klinčić et al., 2020*), especially from indoor environments, so a continued focus on the potential detrimental health effects of this exposure is warranted. In this study, we have shown strong endocrine disrupting effects following developmental exposure to DE-71. In contrast to what was expected from existing studies, however, the endocrine disrupting effects seems to be caused by a complex pattern of modalities and not simply anti-androgenic, as discussed in the following.

Studies on DE-71, including a Hershberger assay, have clearly shown dose-dependent anti-androgenic effects in male rats when exposure occurs during puberty and adulthood. Effects include delayed preputial separation and reduced prostate weights in animals

**Table 3** Absolute organ weights after perinatal exposure to a mixture of brominated flame retardants, Study 2.

| Offspring | Control | 40 mg/kg DE-71 | 60 mg/kg DE-71 |
|---|---|---|---|
| **Male PD16** | $n = 19$ | $n = 18$ | $n = 17$ |
| Body weight, g | 30.6 ± 4.2 | 29.95 ± 5.7 | 28.60 ± 5.1 |
| Ventral prostate, mg | 11.8 ± 4.9 | 9.8 ± 5.1 | **6.8 ± 2.2**[***] |
| **Female PD16** | $n = 21$ | $n = 20$ | $n = 19$ |
| Body weight, g | 30.8 ± 5.0 | 31.0 ± 6.4 | 29.1 ± 5.6 |
| Ovary, right, mg | 2.4 ± 0.6 | 2.3 ± 0.5 | 2.2 ± 0.5 |
| Ovary, left, mg | 2.3 ± 0.6 | 2.3 ± 0.7 | 2.4 ± 0.6 |
| **Male PD300** | $n = 19$ | $n = 18$ | $n = 18$ |
| Body weight, g | 516 ± 69.5 | 549 ± 60.6 | 529 ± 55.6 |
| Prostate and seminal vesicle, g | 3.02 ± 0.38 | 3.14 ± 0.37 | 3.10 ± 0.35 |
| Ventral prostate, g | 0.74 ± 0.12 | 0.69 ± 0.17 | 0.72 ± 0.14 |

**Notes.**
Mean ± SD.
[***]$p < 0.001$ compared to control with body weight as covariate in the analysis.

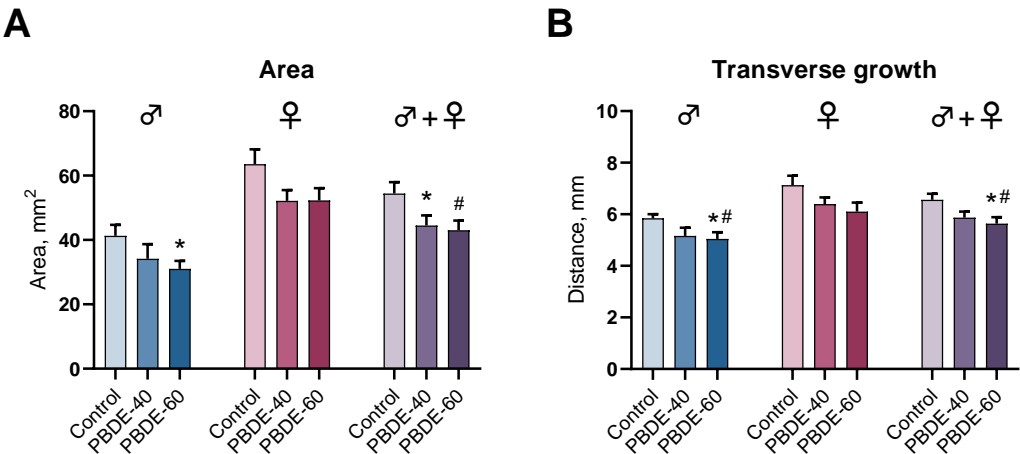

**Figure 4** **Mammary gland development in PD27 offspring after perinatal exposure to brominated flame retardants (DE-71).** (A) Area of the mammary gland was reduced. (B) Transverse growth of the mammary gland was reduced at the highest dose. Mean+SEM. $n = 14$–17 males, $n = 18$–20 females, $n = 32$–37 for males and females pooled and litter effects accounted for in the statistical analysis. Mean+SEM. *$p < 0.05$ (bw as covariate), #$p < 0.05$ using ANOVA.

exposed to 30 mg/kg and above. Additional reproductive tissues such as seminal vesicle, glandula bulbourethralis and penis showed dose-dependent weight reductions at higher doses (up to 240 mg/kg bw/day) (*Stoker et al., 2004*; *Stoker et al., 2005*). Reductions in reproductive organ weights were seen at 200 mg/kg, but not at 67 mg/kg or lower in a 28-day exposure study (*Van der Ven et al., 2008*). In contrast, a dietary study using a 70-day exposure to a mixture of PBDEs (of which 52% was DE-71), but at maximum dose of 20 mg/kg, found no effects on male reproductive organs, hormone concentrations or sperm count (*Ernest et al., 2012*). Thus, PBDEs can seemingly elicit anti-androgenic effects

in pubertal and adult male rats at doses of 30 mg/kg or higher. With respect to fetal and early postnatal exposure, however, there are not much available data.

One study examines effects on anti-androgenic endpoints after developmental exposure (GD21-PND22) to 1, 10 and 30 mg/kg DE-71 administered by gavage (*Kodavanti et al., 2010*). Here effects in the male offspring included a statistically significant 1.8-day delay in preputial separation, a 5% non-significant shortening of AGD, and a 20% non-significant decrease in testosterone concentration on PND 60. Using a similar experimental setup to *Kodavanti et al. (2010)*, but with higher exposure doses, we observed shorter male AGD of around 10% in both dose groups (40 and 60 mg/kg). We also observed a small effect on male nipple retention in the high dose group, with a mean of 0.25 retained nipples—an incidence rate that lies within our historical control data (*Schwartz et al., 2021*) but is significantly different from the concurrent control. Reduced male AGD is associated with reproductive malformations and reduced sperm count and is usually indicative of an anti-androgenic mode of action (*Christiansen et al., 2008*; *Van den Driesche et al., 2011*; *Thankamony et al., 2016*; *Schwartz et al., 2019*). However, we also observed shorter female AGD to a very similar degree as in the male offspring, which we do not normally see for clear anti-androgenic compounds (*Schwartz et al., 2019*). Rather, this effect on female AGD suggests additional modalities induced by the DE-71 mixture.

We have previously shown that developmental exposure to BPA and butylparaben, which mainly are considered estrogenic compounds, albeit they also have other modes of action, induce some of the same effects as those observed with the DE-71 mixture: moderately shorter AGD without dose response, similar effects on AGD in both sexes, and no or minimal effect on nipple retention (*Christiansen et al., 2014*; *Boberg et al., 2016*). This could indicate that some of the same endocrine mechanisms are targeted by BPA, butylparaben and DE-71. However, our understanding of the underlying mechanisms causing this particular effect pattern in males and the reduced AGD in females remain elusive (*Schwartz et al., 2019*). Thus, weak anti-androgenic effects and other endocrine mechanisms could be responsible for the PBDE induced moderate effects on AGD and NR. Notably, intrauterine exposure to PBDEs has been associated with reduced AGD (*Luan et al., 2019*) and increased risk of hypospadias in boys (*Poon et al., 2018*; *Koren et al., 2019*). Although hypospadias has been linked to anti-androgenic chemicals in both humans and rodents, it has also been linked to several compounds with known estrogenic potentials, which suggests the involvement of disrupted androgen-estrogen balance as a critical factor in phallus development (*Mattiske & Pask, 2021*). Taken together, this could suggest that the DE-71 mixture induce a more complex disruption to steroid hormone homeostasis which can result in a varied effect pattern.

The most marked effect observed in the present study was reduction in prostate weights on PD16, which was around 40% decreased in exposed *versus* control animals. Again, this effect could be caused by DE-71 having anti-androgenic properties in the postnatal pups as clearly seen in pubertally exposed animals (*Stoker et al., 2004*; *Stoker et al., 2005*). However, prostate development is also sensitive to estrogen signaling (*Gupta, 2000*; *Prins & Korach, 2008*; *OECD, 2018*) which could also contribute to the observed effect. Interestingly, neither BPA nor butylparaben exposure affected PD16 prostate weights in our studies

(*Boberg et al., 2016*; *Christiansen et al., 2014*) which could indicate that DE-71 works through a different mechanism or possess different ADME (absorption, distribution, metabolism, excretion) properties than BPA and butylparaben in the postnatal period. Overall, it remains possible that the reduced prostate weight induced by DE-71 could have been a result of disrupted androgen-estrogen balance similarly to external genitalia.

In addition to the more commonly assessed effect endpoints AGD, NR and prostate weights, we also assessed mammary gland development in both male and female offspring. DE-71 exposure decreased mammary gland area and transverse growth in perinatally exposed offspring, corroborating findings from a DE-71 toxicity study in Long-Evans rats. In the Long-Evans study only female pups were examined, but perinatally exposed female pups showed less outgrowth, fewer lateral branches and limited terminal end bud (TEB) formation on PND 21 (*Kodavanti et al., 2010*).

The delayed mammary gland development in DE-71 exposed offspring is in contrasts to what would be expected for estrogenic compounds, as accelerated or increased mammary gland growth is typically seen after exposure to estrogenic compounds (*Mandrup et al., 2012*; *Mandrup et al., 2015*; *Macon & Fenton, 2013*) including for BPA and butylparaben (*Boberg et al., 2016*; *Mandrup et al., 2016*). Thus the effects of DE-71 on the mammary glands do not appear to support an estrogenic mode of action. The reduced mammary gland outgrowth caused by DE-71 is also not indicative of anti-androgenic mode of action, as such compounds may not induce an effect on mammary growth in prepubertal stages, but only later in life (*Škarda, 2003*; *Peters et al., 2011*; *Jacobsen et al., 2012*; *Mandrup et al., 2015*). In fact, the observed reductions in mammary gland outgrowth in the present study may best be attributed to an anti-estrogenic mode of action. This fits well with other studies where anti-estrogenic compounds such as ICI 182,780 can cause decreased mammary gland growth and branching in prepubertal female offspring (*Silberstein et al., 1994*; *Cotroneo, 2002*).

Although mammary gland development is sensitive to estrogen signaling, it is also regulated through other signaling pathways. For instance, 2,3,7,8-Tetrachlorodibenzo-p-dioxin (TCDD) given on GD15 can stunt mammary gland growth from PD 4 and persist until PD 68 when the glands still retained undifferentiated terminal structures (*Fenton et al., 2002*). This effect has been suggested to be mediated through AhR activation (*Hushka, Williams & Greenlee, 1998*; *Fenton et al., 2002*; *Helle et al., 2016*), a mechanism that may also be relevant for DE-71 (*Hamers et al., 2006*). Also 3,3′,4,4′,5-pentachlorobiphenyl (PCB-126), which can bind AhR and have anti-estrogenic properties has affected mammary glands similarly to DE-71 (*Fenton et al., 2002*; *Muto et al., 2002*). Thus various environmental chemicals, including DE-71, seem to cause similar effect patterns on the mammary gland albeit the exact mechanism(s) are still unclear.

In conclusion, we have shown that developmental exposure to DE-71 induces endocrine disrupting effects in rats. We found effects on NR and prostate weights in the male offspring, and effects on AGD and mammary gland development in both males and females. The effect pattern is complex and likely involve various mechanisms of action, which are difficult to pin down. The effect-pattern observed in this study suggest that the mixture acts by anti-androgenic, anti-estrogenic and maybe estrogenic modes of action, possibly

in combination with effects on other signaling pathways such as AhR signaling. DE-71 exposure causes endocrine disruption in both male and female rat offspring and thus raises concerns for the long-term consequences of human exposure to PBDEs, especially since associations between PBDE concentrations, AGD, and hypospadias in boys have been reported.

## ACKNOWLEDGEMENTS

We would like to thank Mette Voigt Jessen, Heidi BroksøLetting, Lillian Sztuk, Dorte Lykkegaard Korsbech, Ulla El-Baroudy and Sarah Grundt Simonsen for invaluable technical assistance. We also thank Anne Ørngren and staff for animal care and husbandry.

### Funding
This work was funded by the Danish Environmental Protection Agency, Ministry of Environment and Food of Denmark. The funders had no role in study design, data collection and analysis, decision to publish, or preparation of the manuscript.

### Grant Disclosures
The following grant information was disclosed by the authors:
The Danish Environmental Protection Agency, Ministry of Environment and Food of Denmark.

### Competing Interests
Terje Svingen is an Academic Editor for PeerJ.

### Author Contributions
- Louise Ramhøj conceived and designed the experiments, performed the experiments, analyzed the data, prepared figures and/or tables, authored or reviewed drafts of the paper, and approved the final draft.
- Karen Mandrup conceived and designed the experiments, performed the experiments, analyzed the data, authored or reviewed drafts of the paper, and approved the final draft.
- Ulla Hass and Marta Axelstad conceived and designed the experiments, analyzed the data, authored or reviewed drafts of the paper, and approved the final draft.
- Terje Svingen analyzed the data, authored or reviewed drafts of the paper, and approved the final draft.

### Animal Ethics
The following information was supplied relating to ethical approvals (i.e., approving body and any reference numbers):

Ethical approval was obtained from the Danish Animal Experiments Inspectorate, with authorization number 2012-15-2934-00089 C4. The experiments were overseen by the National Food Institute's in-house Animal Welfare Committee for animal care and use. All methods in the study were performed in accordance with relevant guidelines and regulations.

## Data Availability

The raw data are available in the Supplementary File.

## Supplemental Information

Supplemental information for this article can be found online at http://dx.doi.org/10.7717/peerj.12738#supplemental-information.

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
