# Peer review of "Developmental exposure to the DE-71 mixture of polybrominated diphenyl ether (PBDE) flame retardants induce a complex pattern of endocrine disrupting effects in rats"

_PeerJ, doi:10.7717/peerj.12738_

## Round 0.1 · original submission · Major Revisions

Please see the comments from the reviewers.

Reviewer 1 ·

Basic reporting

The paper reports interesting findings on the effects on reproductive development by the PBDE mixture DE-71, an important global pollutants.
While the results are clearly reported and thoroughly discussed, there are number of problems with study design and interpretation that the Authors need to clarify before recommending the paper for publication.
In its present form, the study can be considered as a pilot trial, prompting toward further investigation.

Experimental design

The sudy design is straightforward, but there are some questions.
- How the dose levels were selected? The lowest dose level (20 mg/kg bw) seems pretty high in comparison with both expected PBDE exposure and nOAEL/LOAEL in previous experimental studies. For instance, EFSA (already in 2011) concluded that the highest exposure to individual major PBDE congeners never exceeded 0.02 mg/kg bw, 3 orders of magnitude lower that 20 mg/kg) and BMDL10 ranging from 0.012 to 1.7 mg/kg bw (3 to 1 order of magnitude lower than 20 mg/kg bw).
RTHe Authors are therefore invited to carefully justify the relevance of the dose range selected
- The choice of endpoints is sound but limited. I understand that the Authors wanted to assess effects on reproductive development, excluding the thyroid and the liver that are well-recognized PBDE targets. I concur with the Authors that more information is needed on PBDE ffects on reproductive development.
Nevertheless is somehow surprising that no investigation was carried on well-known targets of endocrine disruption, siuch as uterus and testis;, no attempts to perform simple quantitative histology were done with prostate and ovary and no hormone measurements were performed.
Overall the many animals (9-12 per litter as I understand from Table 1) seem to have been underxploited.
The Authors shoulsd explain why few (albeit relevant) endpoints were selected
- Finally, data listed in Table 1 look strange: the average perinatal mortality ranges 20-30% in the Control, 420 and 60 mg/kg bw groups, while the postnatal mortality ranges 15-23% in the same groups. These figures seem very high and in the meanwhile have high SD. Can the Authors explain?

Validity of the findings

The findings obtained in the given experimental conditions are valid, properly analysed and reported.
Nevertheless, the Authors need adress questions on the relevance of findings derivingfrom problems in the study design, in particular
- relevance of selected dose levels
- restricted range of findings.
Additional issues
- Actually, 20 mg/kg bw is the NOAEL of this study (no attempt to derive a BMD). This is pretty high, as PBDE may affect, thyroid (and other tissues, such as liver and thymus, see e,g. DOI: 10.1016/j.fct.2013.02.056) at definitely lower levels. Moreover, most effects are observed at 60 mg/kg bw, i.e., with a concurrent general toxicity (although I agree with the Authors that reproductive effects may not be due to general toxicity).
The conclusion seems that reproductive development is not a sensitive target for PBDE. What's the Authors opinion?
- Figure 4: where are the data for the lowest dose level (the most interesting one)? There seems to be a trend toward reduced mammary parameters in females , albeit not significant. Is it the case to attempt a BMD estinate (with values from the 20 mg/kg group, of course)?
What is the rationale to combine gfemales and males as their endocrine regulation is different.
- The two most sensitive endpoints (detected also at the intermediate dose level) are reduced AGD in both sexes and reduced mammary growth parameters (only a trend but in both sexs). These effects fit in a overall picture of general "hormone-inhibiting"effect on both sexes.
In the discussion the Authors do not mention the interactions of PBDE with PXR and CoAR; could these explain the observed effects?
Also thyroid has a modulating effect on reproductive development at early life stages (see e.g., DOI: 10.3389/fendo.2021.736505. Overall, the Authors are invited to discuss the overall picture of effects observed in light of mechanisms other than estrogenicity/androgenicity.

Additional comments

In its present form, the study can be considered as a pilot trial, prompting toward further investigation, pending the replies to the doubts raised above.

Reviewer 2 ·

Basic reporting

Generally speaking, this manuscript is clearly written in professional English and of good format.

Experimental design

PBDEs are widely used flame retardants in daily life and industry and have been demonstrated to harm human and animals' health. Though using PBDEs have been restricted since 2019, humans are still facing exposures to PBDEs in the environment, both indoor and outdoor. This study uses DE71, a mixture of PBDEs, to investigate endocrine disruption effects of developmental exposure to PBDEs in rats. After determining doses of DE71 used in the experiments, anogenital distance, nipple retention, relative prostate weight, mammary gland development were measured in the offspring after perinatal exposure to DE-71, of differential treatment period, respectively.

Validity of the findings

This study has demonstrated that developmental exposure to DE-71causes endocrine disruptive effects in both male and female rats of complex mechanisms to be unveiled. In general, their experimental results support the findings.

Additional comments

There are minor points I’d like to mention. 1) In Line 306, the sentence "..., so too have several compounds with known estrogenic potentials, ..." is confusing, please double check. 2) In Line 318, "mechanismor" should be "mechanism or" with a space in between.

Reviewer 3 ·

Basic reporting

Data from 20 mg/kg dose group is missing.

Experimental design

Authors must be more detailed/specific about what new information was provided in their work.

Validity of the findings

Data in one table does not show asterisks that are indicated in the caption.

Annotated reviews are not available for download in order to protect the identity of reviewers who chose to remain anonymous.

---

## Round 0.2 · accepted · Accept

The reviewer indicated that there was no concern about your revision.

Reviewer 1 ·

Basic reporting

The Authors replied to comments in an adequate way. The revised paper can be accepted for publication

Experimental design

he Authors replied to comments in an adequate way. The revised paper can be accepted for publication

Validity of the findings

he Authors replied to comments in an adequate way. The revised paper can be accepted for publication

Additional comments

No additional comments